# Exploring a New Application of Construct Specification Equations (CSEs) and Entropy: A Pilot Study with Balance Measurements

**DOI:** 10.3390/e25060940

**Published:** 2023-06-15

**Authors:** Jeanette Melin, Helena Fridberg, Eva Ekvall Hansson, Daniel Smedberg, Leslie Pendrill

**Affiliations:** 1Measurement Science and Technology Unit, Division of Safety and Transport, RISE Research Institutes of Sweden, 41258 Gothenburg, Sweden; leslie.pendrill@ri.se; 2Department of Leadership, Demand and Control, Swedish Defence University, 65340 Karlstad, Sweden; 3Community Medicine and Rehabilitation, Physiotherapy, Umeå University, 90187 Umeå, Sweden; helena.fridberg@umu.se; 4Department of Health Sciences, Lund University, 22100 Lund, Sweden; eva.ekvall_hansson@med.lu.se; 5Division of Geriatric Medicine, Skåne University Hospital, Jan Waldenströms gata 35, 20502 Malmö, Sweden

**Keywords:** task difficulty, metrology, validation, Rasch model

## Abstract

Both construct specification equations (CSEs) and entropy can be used to provide a specific, causal, and rigorously mathematical conceptualization of item attributes in order to provide fit-for-purpose measurements of person abilities. This has been previously demonstrated for memory measurements. It can also be reasonably expected to be applicable to other kinds of measures of human abilities and task difficulty in health care, but further exploration is needed about how to incorporate qualitative explanatory variables in the CSE formulation. In this paper we report two case studies exploring the possibilities of advancing CSE and entropy to include human functional balance measurements. In case study I, physiotherapists have formulated a CSE for balance task difficulty by principal component regression of empirical balance task difficulty values from Berg’s Balance Scale transformed using the Rasch model. In case study II, four balance tasks of increasing difficulty due to diminishing bases of support and vision were briefly investigated in relation to entropy as a measure of the amount of information and order as well as physical thermodynamics. The pilot study has explored both methodological and conceptual possibilities and concerns to be considered in further work. The results should not be considered as fully comprehensive or absolute, but rather open up for further discussion and investigations to advance measurements of person balance ability in clinical practice, research, and trials.

## 1. Introduction

There is a global and rapidly increasing interest in the recognising the significance of person-centred care, which includes the use of person-centred outcome measures [1]. Typically, these measures focus directly on the patient’s health and quality of life [2], which are especially relevant alongside the biological signs traditionally in focus in medicine [3]. Thus, health care must manage to measure person attributes such as the constructs of pain, fatigue, independence, balance, mobility, continence, and cognitive capacities [4]. As will be described further in Section 2.1, there is an attribute coupled to person ability; namely, task difficulty [5]. The clinician is typically interested in the person attributes when setting diagnoses, monitoring treatments, evaluating drugs, or understanding disease progression. However, for researchers, metrologists, psychometricians, etc., both coupling attributes [5]—i.e., item and person attributes—are of great importance in much the same way as in a classic weighing where both the mass of the weight and the sensitivity of the instrument used to weigh it both need to be evaluated.

By being able to create and understand and, in turn, claim the highest level of validity of any construct (attributed to items or persons), a construct specification equation (CSE) provides a more specific, causal, and rigorously mathematical conceptualization than any other kind of construct theory [6]. As such, CSEs can guide the formation of an item bank, that is, a set of metrological references for traceable calibration of task difficulty and person ability. The concept of CSE was introduced by Stenner and colleagues in the 1980s [7,8] for a cognitive test—*the Knox Cube Test*—providing an explanation of task difficulty for each of a series of sequences of increasing difficulty estimated after transformation of the raw test scores with the Rasch psychometric model [9]. Today, Stenner’s concept is the key metric behind the reading test Lexile-framework [10]. At the same time, a comparable effort towards CSE formulation in health care has been slow to develop [11,12]. This might be explained by the nature of tests in several parts of health care where there are less pronounced item hierarchies and variations in task difficulty which are, as of yet, only explained in qualitative terms. Another explanation of the lack of development of CSEs in health care seems reasonably to be that the validity concept has been handled to date predominantly with classical test theory—where no proper separation between person and item attributes has been made—and the fact that little attention has been paid to distinguishing between the attribute itself (cf. quantity) from the measured attribute (cf. quantity as measured) [13].

Some related efforts in developing CSEs in other parts of health care have been made, including work in Fisher’s predictive models for physical functioning [14] and Adroher and Tennant’s construct validation of the Evaluation of Daily Activity Questionnaire [15]. In our own work, we have recently started a re-appraisal of CSE in short term memory tests [6,13,16,17,18,19,20], which also include promoting CSE more generally as a kind of ‘recipe’ for certified reference measurement procedures analogous to those in chemical and materials metrology [13,20,21]. Our work has involved quantitative explanatory variables (including task structure parameters such as the number of symbols to be recalled and above all informational entropy). It is not immediately obvious which of these explanatory variables are relevant or applicable to other kinds of tests of human abilities in health care, such as balance measurements. An interesting research question is whether the concept of entropy, which we have found to be a dominant explanatory variable in cognitive tests, is also applicable to other tests in health care. Therefore, our methods need to be extended to be able to fully understand and implement CSEs for constructs in other health care applications.

CSEs formulated with qualitative explanatory variables can also be performed, provided Rasch transformation is done [22]. Earlier, we have tested how a CSE could be obtained with qualitative explanatory variables for a measure of patient experiences of participating in care and rehabilitation [23], which in part corresponds to the method presented by Adroher and Tennant [15]. Here we build further on those methodologies. Specifically, to successfully advance CSE into new fields—such as balance task difficulty—co-creation with people having domain expertise is essential and, we propose four steps (further described in Section 2.3 and Section 2.4): (i) Identification and definition of explanatory variables which increase or decrease linearly with the dependent variable, such as for balance function task difficulty. In this first step, it is important to define the demands required for the tasks themselves, not for a specific person/group of persons performing them. (ii) Scoring of explanatory variables by professionals. At this step we would stress the importance of domain-specific competence for scoring the explanatory variables for each item. (iii) Rasch analysis, as done in Ref. [22], was done to transform of the qualitative explanatory variables prior to formulating CSEs. (iv) A principal component regression (PCR), in order to formulate the CSE [6,13]. In this final step, the full set of identified explanatory variables is evaluated iteratively with evaluations of how model fit changes when explanatory variables are varied. In turn, we seek the highest degree of correlation between empirical task difficulty values and the predicted *Z* as well as smallest measurement uncertainties in the β-coefficients of the CSE (Equation (2), as given in Section 2.4) [17].

Entropy is a broad concept, found in both physics (thermodynamics) and in information theory (Shannon). Different formulations of the concept can have broad application in healthcare, ranging from information studies of cognitive-related aspects [24,25,26] to bodily tasks [27,28,29] and physiological and emotional thermodynamic aspects [30,31] as well as when modelling a measurement process [13]. To explain short-term memory task difficulty in several different recall tasks (blocks, digits, words), an information theoretical approach, particularly extending the deployment of the classic Brillouin entropy expression, has been demonstrated in our work, based on the premise that a task becomes easier when it exhibits more order, and entropy serves as a measure of the amount of order. On this premise, we have shown that the single explanatory variable entropy (as a measure of order) is a dominate factor task difficulty in memory tests with Pearson correlation coefficients of 0.98 [6].

Moving from measurements of human abilities related to *Body function and structures* to observation protocols and questionaries for measurements at the *Activities* and *Participation* level typically involves tasks with subtasks of both physical and psychological actions. Thus, tasks as test items could be determined by breaking them down into their constituent parts [32] and others have suggested that an entropy-related concept, complexity, is critical to understanding such task difficulties. For instance, Commons et al. [33] who have focused on behavioural tasks, proposed that *the higher the order of hierarchical complexity, the more difficult the task*. On the contrary, Adroher and Tennant [15] showed a negative impact of sequence complexity in the context of activities of daily living, which implies that *the higher the number of steps to be performed, the easier the task*. Adroher and Tennant [15] did, however, not specify a definition of sequence complexity. In contrast to our own work, none of this research included considerations of entropy as a measure of order in relation to complexity.

To summarise, we have previously argued both for the role of CSEs and of (informational) entropy as a dominant explanatory variable when ensuring validity in memory measurements and explaining task difficulty [6,16,17]. This should also reasonably apply to other kinds of human abilities and tasks in health care. It, however, warrants exploration of both how to incorporate qualitative explanatory variables in the CSE formulation as well as a better understanding of the order–complexity relationship. In this paper, we report from two case studies. First, in a case study we have explored how the CSE method, including sequence complexity as an independent variable, can be applied to explaining task difficulty in balance measurements. The relation between entropy and complexity is also of interest in that context. Thus, in a second case study, we further investigate the relationship between entropy as a measure of order in balance tasks. This pilot study closes with a discussion of the results of the case studies together and provide recommended further steps as well as possible clinical implications.

## 2. Materials and Methods

### 2.1. Measurements of Human Abilities

In contrast to measurements of physical attributes associated with a person (e.g., length or weight), measures of person attributes such as pain, fatigue, independence, balance, mobility, continence, and cognitive capacities are so-called latent traits. As summarized by Tesio et al. [4]: “*(1) the person’s variable is assumed to be “hidden/latent” within the person; (2) it can be observed only through a sample of potentially infinite behaviors. Therefore, based on observed scores, (3) inferences are necessary to estimate its amount”.* The third inferential step corresponds well with the measurand restitution (denoted “*zR*”) [34]—well-known to physicists and metrologists—where the Rasch formula [9] transforms the observed scores into separate measures of the person and task item attributes, according to the well-known principle of specific objectivity:(1)$$P\left(z_{ij}|\theta_{i},\delta_{j}\right)=\frac{e^{z_{ij}\left(\theta_{i}-\delta_{j}\right)}}{1+e^{z_{ij}\left(\theta_{i}-\delta_{j}\right)}}$$
where $\theta_{i}$ is the latent trait attributed to the person, and $\delta_{j}$ is the latent trait attributed to the tasks.

Observations (e.g., in a balance test or cognitive test) or self-reports (e.g., surveys on pain, fatigue, or quality of life) are two common ways of observing tasks or behaviours to be restituted into latent traits. Which method to be used depends typically on the latent trait to be measured. For instance, in the present case we are using an observation-based balance measurement—Berg’s Balance Scale (BBS)—that takes place in a clinical setting. When using BBS, the test person is asked to perform a set of balance tasks and the test leader (typically a physiotherapist) rates how well he or she performs on a 0 to 4 Likert scale, where a higher rate implies a better performance on each balance task. It should, however, be mentioned that BBS does not provide a measure of the person’s balance ability in ‘the real environment’ nor his or her perceived balance. Similarly, one is using tests clinically to measure cognition, other physical capacities, etc. Thus, it was deemed as a logical step to use an observation-based balance test in a clinic for this study in order to extend our previous methodological work in the development of CSEs for short-term memory, which was also based on clinical tests as opposed to everyday memory ability [35].

Furthermore, for balance, as well as many other human abilities, there are also of course closely linked physical attributes. Specifically, balance-related examples are the vestibular, visual, and somatosensory system and the coordination between these systems. As will be discussed in Section 4, such measures can ideally be used to casually explain the balance ability of a person.

### 2.2. Balance Measurements and Berg’s Balance Scale (BBS)

Berg’s Balance Scale (BBS), developed in 1992 [36], was designed for older people to measure their ability to maintain balance while performing functional tasks in order to monitor balance, to screen rehabilitation needs, and to predict falls. BBS comprises 14 balance tasks that are typical components in everyday life, with the tasks varying by diminishing a person’s base of support and challenging the bodily centre of mass.

Psychometric analyses of BBS have mainly used classical test theory, which simply summarize the raw scores. This implies that it is common to only use raw scores in the clinic, but this has general drawbacks in terms of inappropriate handling of ordinal data and not separating person and item attributes. However, to the best of our knowledge, we are aware of two available studies that have used Rasch analyses to evaluate the measurement properties of BBS. We have utilized empirical data for the present study:Kornetti et al. [37] recruited 100 community-dwelling veterans referred for balance deficits to examine the measurement properties of the BBS;La Porta et al. [38] used a clinical sample with 302 observations from patients with a neurologic disease requiring rehabilitation admitted either as inpatients or outpatients to examine the measurement properties of the BBS.

The study of Kornetti et al. [37], published in 2004, proposed directions for improving the rating scale structure for each of the items, which was examined in the study by La Porta et al. [38] some years later. They, however, did not find a satisfactory solution and proposed further response category modifications and the removal of some items. For our study, item task difficulty values and their associated measurement uncertainties could be included for 14 and 12 items, respectively. The items removed from the analyses by La Porta et al. [38] were BBS02: Standing unsupported and BBS03 Sitting unsupported due to lack of invariance.

### 2.3. Case Study I: Definitions, Collection, and Transformation of Explanatory Variables

In a bachelor thesis, physiotherapy students explored potential explanatory variables for the 14 items in BBS and possible scoring of them [39]. Those were further refined, together with clinical experts on balance, revealing a total of 10 potential explanatory variables (see the first table in Section 3.1). The potential explanatory variables were chosen, based on the assumption that there should be some general demands on the body to perform different tasks, as distinct from explaining an individual person’s ability. To make the scoring as understandable as possible, a pragmatic choice was to use four similar response options to all explanatory variables from no demands/complexity, little demands/complexity, moderate demands/complexity, or large demands/complexity.

In the following step, the scoring of all 11 explanatory variables for all 14 BBS items was done by 3 physiotherapists with specialist competence in balance. They were asked to rate the explanatory variables for the best performance on each item (i.e., a scoring of 4). The scores given for the explanatory variables were then Rasch transformed, following a similar approach as described in Stenner’s original work [7,8]—see the *Introduction*.

### 2.4. Case Study I: Formulation and Evaluation of the CSE

After completing all the pre-processing steps of the data, the fourth step (iv) described above in the Introduction is about the final formulation of the CSE. Typically, the CSE for the dependent variable, **Z**, (e.g., task difficulty, *δ*) is defined as a linear combination of a set, ***k***, (*j* = 1, …, *K*) of explanatory (independent) variables, **X**:(2)$$Z=\overset{K}{\underset{k=1}{\sum }}\beta_{k}\cdot X_{k}$$

As presented in detail elsewhere [13], a CSE can be formulated using a Principal Component Regression (PCR) where a Principal Component Analysis (PCA) amongst the set of explanatory variables, ***X_k,_*** is conducted, followed by a linear regression of the formula attribute values *δ_j_* against ***X’ = X*** × ***P*** in terms of the principal components before a final conversion back from principal components to the explanatory variables, ***X_k,_***. PCR deals with cases where the explanatory variables may not be the experimentally observed quantities, but rather some combination of these in cases where there is a significant correlation between them. Specifically, applying a PCA allows for identification of the main components of variation by “rotating” in the explanatory-variable space from the experimental dimensions to the principal component dimensions. Thus, when using Principal Components (PC), some combination of the explanatory variables is allowed in cases where there is a significant correlation between them in each PC.

Measurement uncertainties for the attribute values *δ_j_* (estimated as *zR* by restitution [34] with the Rasch formula [9])—here balance task difficulty from the two empirical data from the two previously published studies by Kornetti et al. [37] and La Porta et al. [38]—together with the contribution from the explanatory variables will propagate through the PCR to the 𝛽-coefficients in the CSE to the *zR* [17]:(3)$$u\left(zR\right)=\sqrt{\underset{k}{\sum }u{\left(\beta_{k}\right)}^{2}\cdot {\left(x_{k}\right)}^{2}}$$

The performance of the CSE itself as well as the amount of contribution from each explanatory variable are then assessed in terms of (a) the strength of correlation between the predicted task difficulty, *Z*, the empirical values for task difficulty, *δ,* as well as (b) the dispersion of the 𝛽-coefficients of the CSE [17]. Furthermore, a truer picture of the strength of correlation can be given by “reducing” the effects measurement uncertainty by estimating a disattenuated correlation coefficient [40]:(4)$$R_{xy}=\frac{r_{xy}}{\sqrt{r_{xx\cdot }r_{yy}}}\;$$

The PCR methodology was chosen as one among a number of alternative methods (such as linear logistic test model (LLTM), partial least squares (PLS), or lasso regression) [16]. A detailed comparison of these different methods is beyond the scope of this pilot study, where a less “hard-line” statistical approach has been adopted at this stage (see below and Section 4).

### 2.5. Case Study II: Data and Design

Due to the fact that bases for support and vision are critical factors when making balance exercises more challenging in rehabilitation, for case study 2, we first used four balance tasks: (a) standing with wide feet, (b) standing with feet together, (c) standing on one-leg, and (d) standing on one-leg and with closed eyes. These balance tasks are expected to become more difficult when progressing from (a) to (d).

This balance task hierarchy is first exemplified and briefly discussed in relation to previous entropy-related work, followed by an exploration linked to the most similar items in BBS (BBS07: standing close feet; BBS013: tandem stance; BBS014: standing on one leg).

## 3. Results

### 3.1. Case Study I: Development of CSEs for Balance Task Difficulty

A first observation in this case study is that the univariate Pearson correlation coefficients between empirical balance task difficulty values from Kornetti et al. [37] and La Porta et al. [38] were 0.72 (disattenuated correlation = 0.35). This indicates that balance task difficulty is somewhat different for those two empirical studies, which could be due to slightly different populations. In turn, this also implies that the set and contribution from explanatory variables to be used in the CSEs may vary.

In order to demonstrate the development of the CSEs for balance task difficulty in the present work, in a first step, the univariate correlations between balance task difficulty retrieved from the empirical studies [37,38] and the potential explanatory variables (Section 2.3) were investigated. As shown in Table 1, *Complexity* had one of the highest univariate correlations against both measures of task difficulty, while *Coordination* and *Demands for power in foot muscles* were stronger when using the task difficulty values from Kornetti et al. [37] and *Head movement* and *Body movement* were stronger—but negative—when using the task difficulty values from La Porta et al. [38]. However, as also shown in Table 1, adjusted R^2^ values show weaker association between empirical balance task difficulty values and most potential explanatory variables and the standard errors of the estimates are large. When interpretating this, one should also remember that La Porta et al. [38] removed two items (BBS02: Standing unsupported and BBS03 Sitting unsupported). Secondly, the current work is only a pilot study, and we have therefore adopted a less “hard-line” statistical approach at this stage (see below and Section 4).

In the next step, the explanatory variables with correlation coefficients of <−0.2 or >−0.2 were chosen to the development of the CSEs, which are presented in Table 2. Measurement uncertainties, given in brackets after each coefficient, derived in the PCR, are expanded uncertainties U=k·u, with coverage factor, *k* = 2, and u, the standard uncertainty according to the international *Guide to the expression of measurement uncertainty* [41] For these equations explaining balance task difficulty, *Complexity* is significant in both cases, but it is clear that most of the other β-coefficients are dominated by measurement uncertainty, which could call into question the significance of their contributions to task difficulty; even those with relatively high Pearson coefficients; Kornetti et al. [37] R = 0.76 and La Porta et al. [38] R = 0.78. After reducing the effects of measurement uncertainties, the disattenuated correlations were 0.34 and 0.19, respectively. The explored CSEs are thus clearly suffering from large measurement uncertainties, which limits the possibilities in this pilot study to make any definitive conclusions about the predictive power and significant explanations of balance task difficulty.

As the overall aim with this study was to explore and demonstrate the methodology—from the definitions, collections, and transformations of explanatory variables to developing CSEs—in a pilot study, we did not proceed with re-evaluating different sets of explanatory variables. The possibilities and limitations with the methodology as well as implications for the exemplified CSEs are further discussed in Section 4.

### 3.2. Case Study II: Exploration of Using Entropy to Explain Balance Task Difficulty

We expect that balance task difficulty will successively increase from (a) standing with wide feet to (b) standing with feet together to (c) standing on one-leg, and to (d) standing on one-leg and with closed eyes, and we therefore used (a) as the primary task to use for comparison in this explorative work. Thus, compared with (a), (b) is the closest to the primary task, which implies a small change in entropy while (d) is the farthest away from the primary task, which implies a greater change in entropy.

Our earlier studies of memory tests used entropy as a concept in information theory when explaining task difficulty and person ability: higher entropy implies a more ordered task, which should be easier, and a more ordered person should have a higher ability [6,17]. Translating this to the four balance tasks (a)–(d) implies that we would expect there to be more order (less entropy) in (a) standing with wide feet compared to (b) standing with feet together, etc. A more ordered balance task makes—in line with the arguments for case study 1—less demands on managing to maintain one’s balance, as (a) standing with wide feet is less challenging for the postural stability than (b) standing with feet together, or vice versa, (b) standing with feet together will imply a higher degree of postural sway. Specifically, the degree of postural sway complexity can be quantified by multiscale informational entropy (MSE) [28,29]. Again, when explaining balance task difficulty, the explanatory variables must assume that there should be some general demands on the body to perform different tasks, which is different from explaining an individual person’s ability. Thus, it is likely that a group of people who all can perform all four balance tasks equally well, will have a very low variation in MSE for the different tasks and the average MSE should increase linearly as the difficult of the tasks increases from (a) to (d).

Furthermore, with entropy (H(Z) in Equation (5)) as a measure of the degree of disorder, will increase in proportion to the number of competing combinations, *G*!, of a number, *G*, of symbols repeated *N_j_* times, (*j* = 1,…, *M*) for the *j*th symbol (*K* is a normalisation coefficient), according to Brillouin based on the classic Shannon “surprisal” entropy for a probability P:(5)$$H\left(Z\right)=K\cdot lnP=K\cdot \left[ln\left(G!\right)-\overset{M}{\underset{j=1}{\sum }}ln\left(N_{j}!\right)\right]$$

We have previously used this expression to explain task difficulty in memory tests [6,13,16,17,18,19,20]—see the Introduction. Entropy is a broad concept, including both the amount of information, degree of order, as well as uncertainty [5]. Higher disorder leads to more uncertainty as studied by Hirsh et al. [24]—again based on Shannon’s formula—in terms of uncertainty-related anxiety where they expressed the amount of uncertainty as entropy, claiming—as above—that it *will increase in proportion to the number of competing possibilities that it must be selected from*. Thus, situations with a large range of perceived possibilities imply greater uncertainty expressed as entropy, and on the contrary, situations with less possibilities will result in states of relatively less uncertainty and reduced entropy [24]. Translating this to the four balance tasks (a)–(d), the difficulty of the tasks decreases linearly where those most familiar in daily life are the easiest tasks. Specifically, it is much more common—i.e., there are fewer competing possibilities to choose from—when (a) standing with wide feet compared to (b) standing with feet together, followed by less common to (c) standing on one-leg, and to (d) standing on one-leg and with closed eyes.

In the next step, we relate this proposal of entropy as a measure of the amount of information and order to the results of case study I and the role of *Complexity* as an explanatory variable for balance task difficulty. Table 3 shows the empirical task difficulty values from Kornetti et al. [37], La Porta et al. [38], and *Complexity* for the most similar items in BBS (BBS07, BBS013, and BBS014) to our four balance tasks. The first note to be made is the significant difference in BBS013: tandem stance between the two empirical studies; specifically, in Kornetti et al. [37] it was significantly more difficult than BBS014: *standing on one leg*, while these items cannot be separated in the study by La Porta et al. [38]. Complexity, on the other hand, followed more empirical values from La Porta et al. [38] using the same values for BBS013: tandem stance and BBS014: standing on one leg. Bringing this together, the proposed steps forward are to study how postural sway complexity can be best quantified by MSE and how competing possibilities can be quantified with Shannon’s formula. These approaches are related to the very pragmatic rating and poorly defined complexity in case study 1, aiming to further investigate the understanding of the order–complexity relationship and explain the task difficulty of balance tasks.

A third example is to complement entropy as a concept in information theory and draw analogies additionally with physical thermodynamic entropy—which is a related, but distinct concept to information entropy. For instance, Boregowada et al. [30] used a ‘relaxation period’ where several human physiological responses were compared in order to calculate two types of thermodynamic entropy change when performing different cognitive tasks. They considered the two subsystems in the human physiology to correspond to closed thermodynamic systems, and the two measures of entropy change were Δ𝑆BP due to blood pressure, heart rate, and skin temperature and Δ𝑆EMG due to electromyogram, electrodermal response, and skin temperature [30]. Based on those physiological responses, Δ𝑆EMG is somewhat related to the explanatory variables used in the case study I: *Body movement* and *Demands for power in feet, knee, hip, and back muscles*. Such thermodynamic entropy terms are arguably more characteristic of the response of a living being as opposed to informational entropy terms, which could equally apply to an inanimate robot.

Following this proposal, including similar items from BBS to our (a)–(d) balance tasks, Table 4 shows a type of heat-map for five potential explanatory variables from case study I related to the physiological thermodynamic entropy change in the study of Boregowada et al. [30]. The balance tasks are ordered as they appear in BBS, which corresponds well with the empirical task difficulty values from La Porta et al. [38]. Measurement uncertainties are in most cases too large to truly separate out variation in task difficulty due to the explanatory variables. There seems to be some indication of a linear increase of explanatory variables with respect to *Demands for power in feet, knee, and hip muscles*. Thus, further investigations of such demands in relation to the electromyogram as a generic demand on the body (similarly as exemplified with MSE) are of of particular interest.

## 4. Discussion and Conclusions

The results of the first of the two case studies in this paper have demonstrated the challenges to be faced when attempting to extend the CSE methodology to qualitative explanatory variables. The results of the second case study have opened up the possibilities for the introduction of more rigorous, entropy-based explanatory variables to balance studies.

Although explanations of the differences in empirical balance task difficulty values between Kornetti et al. [37] and La Porta et al. [38] noted in case study I are beyond the scope of this study, they do have implications when seeking to explain a unidimensional continuum ranging from the easiest to the most difficult items in a coherent and consistent way. Having different CSEs—including different explanatory variables—raises questions about the construct validity of balance task difficulty and in turn challenges the invariance and comparability of measures of a person’s balance ability. However, measurement uncertainties are found to be large in our study and the clinical implications of the differences warrant further investigations. Case study II provides three alternative routes to further investigate entropy in relation to balance tasks. We encourage exploration of such variables in CSEs in order to advance the pragmatic decisions made in case study I. One cannot, however, ignore the fact that empirical balance task difficulty values from different cohorts are different, and such differences need to be investigated with larger and more diverse samples to understand the potential variations due to sample-related factors.

The separation between person and item attributes is necessary for measurement quality assurance of human-based measurements, such as balance measurements. While there is a relation between them—a purely mathematical relation—this is not the same as the stricter kind of causality captured by the CSE [17]. In this paper we have focused on explaining balance task difficulty, while the same methodology could of course be applied to explain person balance ability. For instance, we have proposed the possibilities to include entropy-based terms for connectivity and Functional Magnetic Resonance Imaging (fMRI) to explain person memory ability [6,17]. Recent work by Becker and Hung [26] about balance ability in terms of sampling entropy, has demonstrated that an external focus is superior to a holistic focus and an internal focus for persons when adapting to movement adjustments. Such methods might be useful when investigating other possible compensatory strategies that can be used either to differentiate persons along the balance ability continuum or to identify sub-sample variations. Likewise, there are also of course other balance-related physical attributes, such as vestibular, visual, and somatosensory variables to be used to causally explain person balance ability.

The proposed methods in case study II to causally explain person balance ability can also be followed up. As an example, as stated above, it is likely that a group of people who can all perform all four balance tasks equally well will have a very low variation in MSE for the different tasks and the average MSE will increase linearly as the difficult of the tasks increases from (a) to (d). At the same time, for a group of people with varying balance ability, it is likely that they will exhibit variations in MSE for a single balance task, which might be used to explain each individual’s balance ability.

A key part of this study was also to investigate the methodological possibilities when developing CSEs with qualitative explanatory variables. Based on case study I, the four steps outlined in the introduction are a useful point of departure for structuring this work. However, as we aim to introduce new and innovative ideas, some pragmatical decisions are additionally needed and revisited. Specifically, we would encourage the evaluation of a pragmatic balance between the level of detail and specificity of the explanatory variables. Additionally, we recommend recruiting more clinicians to provide ratings, especially considering the large uncertainties in CSEs found in this pilot study. We believe that such methodological advancements parallel to the exploration of possibilities to quantify postural sway complexity, competing possibilities, Δ𝑆EMG, etc., for balance task difficulty can enrich the CSE as a methodology as well as contribute to a better understanding of the role of entropy and order–complexity relationship.

Furthermore, Green and Smith [42] highlight four concerns—the effects of sample size, collinearity, measurement disturbance, and multidimensionality—when formulating CSEs. Large and well targeted samples provide more information about each item—here balance task—and when the sample size increases, the measurement uncertainties for estimates of task difficulty can be reduced [43]. In case study 1, we used empirical data from previously published studies with rather small sample sizes, viz., 100 [37] and 302 [38]. Although initial estimates of the reliability in the empirical task difficulties were rather high (0.91 and 0.83), disattenuation for uncertainty resulted in significantly decreased correlations. It should be noted that the disattenuation is not a shortcut to more precise measurements or improving the measurement quality; rather, it clarifies the impact from measurement uncertainties [44]. Our results could be interpreted as showing significant effects of measurement disturbance among the explanatory variables, which is also confirmed by the large measurement uncertainties in the 𝛽-coefficients we declared. This calls for further investigation of the role of one or more additional explanatory variables associated with unrecognized and unexplained variation, as explored in case study 2. These issues could also be signs of collinearity and other disturbances, which warrants further investigation [17,42] of the present pilot study. Collinearity should be handled in principle by the PCR methodology. Specifically, the first step with the PCA allows for identification of the main components of variation by “rotating” in the explanatory-variable space from the experimental dimensions to the principal component dimensions. A detailed comparison of the PCR compared to other methods, such as LLTM or PLS, is recommended. The last point, multidimensionality, which always exists to some extent [45], simultaneously raises the question of whether it is significant enough to maintain a strict unidimensional model or not [16]. When comparing the set of items used, it is worth noting that La Porta et al. [38] had to remove two items due to item misfit. Additionally, the correlations between the empirical balance task difficulty values raise questions about construct validity, as mentioned earlier. Thus, with less well-defined or well-constructed items, it is less likely that an appropriate CSE model would be identified [42]. In case study 1, different explanatory variables were allowed, but with a stricter view on unidimensional we would expect that the same underlying variables would explain the construct [42].

In the present pilot work, various potential routes forward to CSE methodology to enhance measurement quality assurance in balance measurements have been indicated. There remain some methodological limitations to consider when interpretating the results. First, we have used results from two studies showing an inconsistent item hierarchy, and in turn, one can questioning the validity in balance task difficulty as an invariant attribute. Secondly, only three physiotherapists made the ratings of the explanatory variables in case study 1. They, however, possessed specialist competence in balance, which was deemed more important than the number of raters. Thirdly, entropy was not used as an explanatory variable in case study 1 since this was a term unfamiliar to clinicians. Finally, case study 2 is yet only a discussion and does not include empirical data for our proposals.

To conclude, this pilot study has explored both methodological and conceptual possibilities. The results in this paper should not be considered as fully comprehensive or absolute, but rather opening for further discussion and investigations to advance measurements of a person’s balance ability in clinical practice, research, and trials.

## Figures and Tables

**Table 1 entropy-25-00940-t001:** Pearson correlation coefficients between task difficulty values from Kornetti et al. [37] and La Porta et al. [38] against all 11 potential explanatory variables (Section 2.3).

Explanatory Variables	Kornetti et al. [37]	La Porta et al. [38]
Demands for head movement	0.07	−0.28 *
Demands for body movement	−0.02	−0.35 *
Demands for range of movement feet	0.03	−0.17
Demands for range of movement knees	−0.04	−0.19
Demands for range of movement hips	0.00	−0.06
Demands power in foot muscles	0.40 *	0.13
Demands power in knee muscles	0.14	−0.25 *
Demands power in hip muscles	0.24 *	−0.14
Demands power in back muscles	−0.18	−0.23 *
Demands for coordination	0.42 *	0.17
Complexity	0.56 *	0.30 *

* Correlation coefficients <−0.2 or >0.2 indicate variables that were proceeded with into the CSEs.

**Table 2 entropy-25-00940-t002:** Examples of CSEs for balance task difficulty for empirical values from Kornetti et al. [37] and La Porta et al. [38].

Explanatory Variables	Kornetti et al. [37]	La Porta et al. [38]
Intercept	−1.19 (1.68)	−1.99 (2.62)
Demands for head movement		−0.14 (46)
Demands for body movement		−0.05 (17)
Demands power in foot muscles	0.74 (1.10)	
Demands power in knee muscles		−0.12 (31)
Demands power in hip muscles	−0.44 (73)	
Demands power in back muscles		−0.08 (11)
Demands for coordination	−0.27 (20)	
Complexity	0.54 (52)	0.28 (19)

**Table 3 entropy-25-00940-t003:** Empirical balance task difficulty values for three selected items from BBS from Kornetti et al. [37] and La Porta et al. [38] as well as measurement value for *Complexity* from case study 1. The numbers in brackets are *k* = 2.

Balance Task	Kornetti et al. [37]	La Porta et al. [38]	Complexity
BBS07: standing close feet	1.6 (0.40)	−0.49 (0.30)	−0.06 (2.8)
BBS013: tandem stance	5.7 (0.52)	1.81 (0.29)	10.08 (12.7)
BBS014: standing on one leg	2.4 (0.38)	1.99 (0.30)	10.08 (12.7)

**Table 4 entropy-25-00940-t004:** Empirical balance task difficulty values for three selected items from BBS from Kornetti et al. [37] and La Porta et al. [38] as well as measurement values for *Demands for power in feet, knee, hip, and back muscles* from case study 1. The numbers in brackets are *k* = 2.

Balance Task	Kornetti et al. [37]	La Porta et al. [38]	Body Movement	Demands for Power in Feet Muscles	Demands for Power in Knees Muscles	Demands for Power in Hips Muscles	Demands for Power in Back Muscles
BBS07: standing close feet	1.6 (0.40)	−0.49 (0.30)	−4.35 (7.66)	−0.13 (2.22)	−1.75 (2.78)	0.41 (3.02)	−2.59 (2.66)
BBS013: tandem stance	5.7 (0.52)	1.81 (0.29)	−6.02 (17.38)	4.05 (2.34)	3.9 (4.84)	3.87 (5.1)	−6.64 (8.14)
BBS014: standing on one leg	2.4 (0.38)	1.99 (0.30)	−6.02 (17.38)	5.5 (3.78)	9.21 (3.18)	11.27 (4.1)	−0.81 (2.82)

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
