# Peer review of "Exploring a New Application of Construct Specification Equations (CSEs) and Entropy: A Pilot Study with Balance Measurements"

_entropy, 2023, doi:10.3390/e25060940_

Round 1
Reviewer 1 Report
This paper presents a report of two case studies exploring the possibilities of advancing construct specification equations (CSEs) and entropy in balance measurements.
The paper is well organized, and the length is appropriate. The title is chosen correctly, and the abstract provides sufficient information to understand what to expect from the paper.
The references are relevant and correctly chosen, and related work is discussed and cited appropriately.
The study methods are appropriate, and the data are valid.
The results are well highlighted, and the conclusions are adequate.
Minor comments:
1. Table 3 should be modified so that the meaning of the last four columns can be easily read, either by adding separator lines or using a smaller font.
2. In the discussion section, a paragraph should be added on the limitations of the study
Author Response
|
Reviewer comment |
Response |
|
This paper presents a report of two case studies exploring the possibilities of advancing construct specification equations (CSEs) and entropy in balance measurements.
The paper is well organized, and the length is appropriate. The title is chosen correctly, and the abstract provides sufficient information to understand what to expect from the paper.
The references are relevant and correctly chosen, and related work is discussed and cited appropriately.
The study methods are appropriate, and the data are valid.
The results are well highlighted, and the conclusions are adequate. |
We thank you for your appreciated words. |
|
Table 3 should be modified so that the meaning of the last four columns can be easily read, either by adding separator lines or using a smaller font. |
We changed to a smaller font in all tables. |
|
In the discussion section, a paragraph should be added on the limitations of the study |
We have added a section on study limitations. |
Reviewer 2 Report
The submitted manuscript applies two approaches for predicting item difficulties obtained from a Rasch model (RM): the construct specification approach and an entropy approach. Overall, I found that the manuscript must improve in several aspects before being considered for publication.
Major comments:
1. Abstract, l. 21: It is certainly a misrepresentation to state that “linear measures” can be obtained from the RM. In this place (and other places of the manuscript), the authors seem to claim that the RM would provide interval-scale measurements, while raw scores (e.g., sum scores or mean scores) do not. I disagree with such reasoning (e.g., see Feuerstahler, 2023). For example, in l. 90, it stated that “linear interval measures” are derived through the RM (see also 207). This statement is unfounded, in my opinion.
2. The authors should include some brief mathematical details about the RM.
3. 147: What is the “Rasch-formula”?
4. It seems that only 14 tasks (i.e., 14 items) were used for predicting item difficulties with 10 explanatory variables. I think that this is a major drawback of this study. The generalizability should be proven using a larger set of items.
5. The authors use principal component regression (PCR) because the number of variables p is large compared to the number of cases n. In such a situation, I would prefer either partial least squares or lasso regression.
6. 191: Which kind of noninvariance led to the removal of two items?
7. Section 2.4: Include a motivation for why PCR is applied.
8. Equation (1): Define indices for the sum symbol (i.e., “k=1” to “K” or similar) and all symbols appearing in the formula.
9. 240: A correlation of 0.72 cannot be judged unless this correlation is corrected for sampling errors (i.e., estimated item difficulties are affected by sampling errors, which, in turn, reduce the correlation).
10. Include standard errors in Table 1.
11. Report the results of Equations (2) and (3) in a table as it is usual in scientific papers.
12. Report R^2 and adjusted R^2 for the two regression models.
13. 265: What is meant by “measurement uncertainties”? I suppose that standard errors refer to the error in the regression model based on statistical inference for items, not persons. This should be mentioned somewhere.
14. Notably, the linear model for predicting item difficulties is well-known as a linear logistic test model (LLTM) in the RM literature. This should be properly addressed and cited elsewhere in the manuscript.
15. Section 3.1.: Please include mathematical details about the entropy approach. Without reading the original literature, I couldn't get an idea of what the authors had done.
Minor comments:
16. 39: Write “Section 2.1”
17. 117: write “Commons et al.”
18. 131: write “we report results from …”
19. 143: Use the quotation for the cited statement of Tesio. Do not italicize the original quote.
20. 153, 170: Inconsistent use of “Bergs Balance Scale” and “Berg’s Balance Scale”
References:
Feuerstahler, L. Scale Type Revisited: Some Misconceptions, Misinterpretations, and Recommendations. Psych 2023, 5, 234–248. https://doi.org/10.3390/psych5020018
Minor editing of English language required
Author Response
|
Reviewer comment |
Response |
|
1. Abstract, l. 21: It is certainly a misrepresentation to state that “linear measures” can be obtained from the RM. In this place (and other places of the manuscript), the authors seem to claim that the RM would provide interval-scale measurements, while raw scores (e.g., sum scores or mean scores) do not. I disagree with such reasoning (e.g., see Feuerstahler, 2023). For example, in l. 90, it stated that “linear interval measures” are derived through the RM (see also 207). This statement is unfounded, in my opinion. |
In this work, RM is applied in a more or less regular fashion as in earlier works, as noted in the text (in the Abstract and elsewhere in the manuscript - see track changes). Although an interesting discussion could in principle be continued along the lines indicated by Reviewer 2 and by others, it is felt to be somewhat out of the scope of the present work where the research novelty lies elsewhere. |
|
2. The authors should include some brief mathematical details about the RM. |
The mathematical expression of RM has been added on line 155. |
|
3. 147: What is the “Rasch-formula”? |
The mathematical expression for RM is typically referred to as the “Rasch-formula”. |
|
4. It seems that only 14 tasks (i.e., 14 items) were used for predicting item difficulties with 10 explanatory variables. I think that this is a major drawback of this study. The generalizability should be proven using a larger set of items. |
We have added a section on study limitations addressing this. |
|
5. The authors use principal component regression (PCR) because the number of variables p is large compared to the number of cases n. In such a situation, I would prefer either partial least squares or lasso regression. |
We have added a justification for the PCR in lines 237-244. |
|
6. 191: Which kind of non invariance led to the removal of two items? |
As can be read in La Porta et al 2012, both BBS02 and BBS03 showed misfit to the model and therefore they were removed. |
|
7. Section 2.4: Include a motivation for why PCR is applied. |
We have added a justification for the PCR in lines 237-244. |
|
8. Equation (1): Define indices for the sum symbol (i.e., “k=1”to “K” or similar) and all symbols appearing in the formula. |
k has been defined in line 222. |
|
9. 240: A correlation of 0.72 cannot be judged unless this correlation is corrected for sampling errors (i.e., estimated item difficulties are affected by sampling errors, which, in turn, reduce the correlation). |
R^2, adjusted R^2 and std. error have been added.
|
|
10. Include standard errors in Table 1. |
R^2, adjusted R^2 and std. error have been added. |
|
11. Report the results of Equations (2) and (3) in a table as it is usual in scientific papers. |
We have reported in this way previously, for instance in this journal, and think it is the best way to present the results when there are only two equations and especially as they have different explanatory variables. |
|
12. Report R^2 and adjusted R^2 for the two regression models. |
R^2, adjusted R^2 and std. error have been added. |
|
13. 265: What is meant by “measurement uncertainties”? I suppose that standard errors refer to the error in the regression model based on statistical inference for items, not persons. This should be mentioned somewhere. |
Text added. “Measurement uncertainty” is the accepted international terminology – see tracked changes for the added reference. The uncertainties shown are, as stated, those in the given equations 3 & 4: yes, for the items. |
|
14. Notably, the linear model for predicting item difficulties is well-known as a linear logistic test model (LLTM) in the RM literature. This should be properly addressed and cited elsewhere in the manuscript. |
We have added a justification for the PCR in lines 237-244. |
|
15. Section 3.1.: Please include mathematical details about the entropy approach. Without reading the original literature, I couldn't get an idea of what the authors had done. |
The Brillouin expression and Shannon formula have been added line 331. |
|
16. 39: Write “Section 2.1” |
Corrected. |
|
17. 117: write “Commons et al.” |
Corrected. |
|
18. 131: write “we report results from …” |
Revised. |
|
19. 143: Use the quotation for the cited statement of Tesio. Do not italicize the original quote. |
Corrected. |
|
20. 153, 170: Inconsistent use of “Bergs Balance Scale” and“Berg’s Balance Scale” |
Corrected. |
Round 2
Reviewer 2 Report
Several of my comments remained unaddressed or inadequately addressed, e.g., my previous Comment 5 (“240: A correlation of 0.72 cannot be judged unless this correlation is corrected for sampling errors (i.e., estimated item difficulties are affected by sampling errors, which, in turn, reduce the correlation). The large standard errors in model parameters indicate untrustworthy results.
There may be reviewers who are more positive about the paper, but I am not convinced about the scientific content of the paper.
Author Response
Dear Editor and Reviewer 2,
We thank you for inviting us to provide a further revision of our manuscript.
Given diverse scientific traditions, backgrounds, and experiences, researchers will have different views. In our case, it was evident that Reviewer 1 was more supportive of our approach than Reviewer 2. This is also mentioned by reviewer 2 in the latest report “There may be reviewers who are more positive about the paper, but I am not convinced about the scientific content of the paper.”
Our paper was submitted as a pilot study taking an explorative approach. We have intended to draw careful conclusions throughout this work with due respect for the large measurement uncertainties we have openly declared. At the same time, exploring new areas such as in the current work must allow a not-too-hard-line-driven statistical approach but rather open up possibilities for further discussion and investigations in order to advance measurements. In the revised manuscript, we have therefore stressed at several additional places to remind the reader that this is a pilot work.
Unfortunately, reviewer 2 does not feel that we have addressed initial comments adequately or that they are unaddressed. We did our best to respond and apologies if something has been left out or misunderstood. For instance, we added a paragraph about PCR (slightly revised in the revised manuscript, see track-changes) compared with other methods and a limitations section. We also added, as requested, additional information about the Pearson correlation coefficients. Previously, we also have at some places in the manuscript (e.g., lines 247-253 and 392-404) discussed the limitations with empirical studies showing such variations. In order to further address the comments if still felt necessary, we would need to have additional specifications about what the reviewer exactly wants to be developed/revised/removed.
Round 3
Reviewer 2 Report
The revised manuscript applies two approaches for predicting item difficulties obtained from a Rasch model (RM): the construct specification approach and an entropy approach. Here are a few detailed comments for improving the manuscript:
1. L. 275: As I pointed out in my previous review, you have to correct the correlation of balance task difficulties for unreliability (i.e., the presence of standard errors due to sampling of persons). Reporting an adjusted R^2 is by no means a correction for unreliability. Use an appropriate attenuation formula.
2. 275: What is the “Std. error of the estimate”? Which estimate?
3. Equations (3) and (4): Report the equations in a table. The fact that you find a manuscript that also uses equations for reporting provides no justification for using equations again.
4. Eq. (3), (4): Standard errors (measurement uncertainties in your terminology?) were reported. This means that no coefficients are statistically significant. Hence, no valid conclusions can be drawn. Moreover, does uncertainty refers to the number of persons or the number of items? I missed a discussion about which kind of significance (i.e., uncertainty) should be quantified.
5. 30: change the keyword “Rasch model”
Author Response
|
Reviewer comment |
Response |
|
1. L. 275: As I pointed out in my previous review, you have to correct the correlation of balance task difficulties for unreliability (i.e., the presence of standard errors due to sampling of persons). Reporting an adjusted R^2 is by no means a correction for unreliability. Use an appropriate attenuation formula |
Thanks for clarification. We have replaced adjusted R2^2 with disattenuated correlations. We have also added a new section with a more extensive discussion on the role of uncertainties and weaknesses in correlations to the discussion section. |
|
2. 275: What is the “Std. error of the estimate”? Which estimate? |
Per your request on point 1, we have replaced it with the disattenuated correlation. |
|
3. Equations (3) and (4): Report the equations in a table. The fact that you find a manuscript that also uses equations for reporting provides no justification for using equations again |
Previous Eq. 3 and 4 are now presented in new Table 2. |
|
4. Eq. (3), (4): Standard errors (measurement uncertainties in your terminology?) were reported. This means that no coefficients are statistically significant. Hence, no valid conclusions can be drawn. Moreover, does uncertainty refers to the number of persons or the number of items? I missed a discussion about which kind of significance (i.e., uncertainty) should be quantified. |
Complexity is significant in both cases. We have commented on this in text to avoid confusion.
We have added a short description on measurement uncertainties propagates from empirical values in the method and reference to a report elsewhere.
Furthermore, a new section with a more extensive discussion on the role of uncertainties and weaknesses in correlations has been added to the discussion section. |
|
5. 30: change the keyword “Rasch model” |
We have changed from “Rasch” to “Rasch model” |
Round 4
Reviewer 2 Report
no further comments